# Lactation Activity and Mechanism of Milk-Protein Synthesis by Peptides from Oyster Hydrolysates

**DOI:** 10.3390/nu14091786

**Published:** 2022-04-24

**Authors:** Suhua Chen, Xiaoming Qin, Chaohua Zhang, Wenhong Cao, Huina Zheng, Haisheng Lin

**Affiliations:** 1National Research and Development Branch Center for Shellfish Processing (Zhanjiang), Guangdong Provincial Key Laboratory of Aquatic Products Processing and Safety, Guangdong Province Engineering Laboratory for Marine Biological Products, Key Laboratory of Advanced Processing of Aquatic Product of Guangdong Higher Education Institution, College of Food Science and Technology, Guangdong Ocean University, Zhanjiang 524088, China; chensuhua1@stu.gdou.edu.cn (S.C.); zhangch@gdou.edu.cn (C.Z.); cwenhong@gdou.edu.cn (W.C.); margaretphd@126.com (H.Z.); haishenglin@163.com (H.L.); 2Collaborative Innovation Center of Seafood Deep Processing, Dalian Polytechnic University, Dalian 116034, China

**Keywords:** oyster (*Crassostrea hongkongensis*), mammary gland, prolactin, lactation, human mammary epithelial cells, milk-protein synthesis

## Abstract

Oyster meat has a tender texture and delicate flavor, and the oyster is an aquatic shellfish with high nutritional and economic values. As they are rich in protein, oysters serve as a good source for the preparation of bioactive peptides. However, research on the lactation effect and mechanism of the synthesis of polypeptides from oyster hydrolysates is yet to be observed. This study aimed to analyze the lactation activity of the fraction UEC4-1 and explore its mechanism. The results show that, in an in vivo experiment, UEC4-1 could significantly increase the concentration of PRL in the serum and mammary tissue and the concentration of PRLR in the mammary tissue in rats with postpartum hypogalactia. UEC4-1 promoted the development of mammary tissue structure, resulting in active lactation. UEC4-1 promoted the proliferation of MCF-10A in a dose-dependent manner and could significantly upregulate the gene expression levels of PRL, PRLR, CSN1S1, CSN2, CSN3 and CCND1. UEC4-1 could also significantly increase the expression of mTOR, AKT1, RPS6KB1 and STAT5A in MCF-10A and improve its phosphorylation level. These results show that UEC4-1 had the ability to upregulate the proliferation and PRL synthesis of MCF-10A and promote lactation. The ability of UEC4-1 to regulate the milk-protein synthesis signaling pathway is the mechanism behind this. Oysters had a remarkable effect on lactating mothers’ sweating irritability after childbirth and may serve as an everyday diet to promote lactation. Postpartum dysgalactia is a common problem for lactating women. The study of the oyster’s lactation-active peptide can provide dietary nutrition guidance for postpartum lactating mothers, and it has the potential to be used for the development of drugs for the treatment of postpartum hypogalactia or oligogalactia.

## 1. Introduction

Low breast milk supply is the most frequent cause of breastfeeding failure [1]. Identifying safe and efficient dairy products or drugs is an urgent task for scientists. Lactation is defined as the secretion of milk from the mammary gland (MG) and is influenced by a complex hormonal network [2]. Although initially required for the morphological development and differentiation of the MG, prolactin (PRL) plays a crucial role in stimulating milk protein (MP) and lactose synthesis to promote milk synthesis [3,4]. Breastfeeding is arguably the best way to feed. Breast milk can provide a variety of nutritional and bioactive health factors, breastfeeding can improve the health status and immune system development of infants and breastfed infants have a lower incidence of gastrointestinal diseases and mortality [5,6,7]. Amino acids (AAs) play crucial roles in the synthesis of MP in the MG. Most recently, in Wu’s review [8], the amino acid uptake/output ratios in four different mammals showed that five AAs are associated with lactation: Val, Ile, Leu, Arg and His. Among these, in vitro experiments in mammary epithelial cells (MECs) demonstrated that Val could increase β-casein synthesis by activating the mammalian target of rapamycin (mTOR) and the Ras/ERK signaling pathway and Arg could promote cell proliferation and increase protein synthesis by activating the mTOR signaling pathway. However, the underlying mechanisms and signaling pathways by which AAs regulate milk synthesis were largely unknown until recently. Zhou et al. summarized some recent studies on the effects of EAAs on MP synthesis in dairy cows. Essential amino acids (EAAs) that can increase milk yield and MP yield include Met, Val, His, Phe, Thr, Arg and Trp. In addition, Val can increase the transcription of casein’s genes and His can promote β-casein synthesis and increase milk yield, and it tends to increase the MP yield. Moreover, Phe, Thr, Arg and Trp are positively associated with milk yield [9]. These studies on lactation have mainly focused on dairy cows, goats, sows and ewes, with the purpose of improving the quality of dairy products, milk yield and quality. There is still a lack of research on human lactation, which needs to be developed.

The oyster is one of the main shellfish cultured in the coastal areas of Guangdong, Guangxi, Fujian and Shandong. The Hong Kong oyster (*Crassostrea hongkongensis*) is an economically important invertebrate that is found in mudflats and is widely cultivated in southern China [10]. Around 400 years ago, it was recorded in *Ben Cao Gang Mu* that oyster meat cured asthenia and healed women’s blood. Around 200 years ago, it was recorded in Fu Qing-zhu Gynecology that oysters had a remarkable effect on sweating irritability. According to the Chinese Materia Medica, oysters have long been used in foods and pharmaceuticals. As they are very rich in protein, oysters are a good source for the preparation of bioactive peptides.

Studies have shown that the polypeptides from oyster enzymatic hydrolysates have various physiological activities, such as enhancing immunity [11] and antitumor [12,13], anti-inflammatory [14,15] and antioxidative effects [16]. There is no literature related to oyster peptides and lactation. In our previous study, an enzymatic hydrolysate of *Crassostrea hongkongensis* could obviously increase hourly lactation in overloaded lactating rats and pup weight, significantly promoted the expansion and filling of the mammary gland’s acinar cavity in these rats and promoted lactation [17]. This suggests that oysters can be used as a dietary source of exogenous peptides to promote milk secretion. More recently, Cai et al. found that the bioactive polypeptide OPH3-1 of octopus protein hydrolysate can promote the proliferation of mouse mammary epithelial cells and promote the synthesis of β-casein [18]. Other studies found that small peptides can be absorbed in an intact form [19,20,21]. Moreover, peptides appear to be more efficient than free amino acids in promoting protein synthesis in lactating mammary tissue, but little is known about the underlying mechanisms. Thus, further understanding the nutritional strategies with which to regulate milk synthesis and its underlying mechanism is important for human beings.

In this study, we studied the potential efficacy and mechanism of the active peptide UEC4-1 for improving postpartum hypogalactia in rats with hypogalactia. The effect of UEC4-1 on the proliferation of HMECs was studied to lay the foundation for structural analysis and research into the lactation mechanism.

## 2. Materials and Methods

### 2.1. Materials

Live Hong Kong oysters (*Crassostrea hongkongensis*) were obtained from the Dongfeng Aquatic Products Market (Zhanjiang, China).

The normal human mammary epithelial cell line MCF-10A (item: BNCC 341931), a complete growth medium (DMEM/F12 + 5% HS + 20 ng/mL EGF + 0.5 μg/mL hydrocortisone + 10 μg/mL insulin + 1% NEAA + 1% P/S) and DMEM/F12 (1:1) basic (1×) (Gibco, C11330500BT) were purchased from Beina Chuanglian Biotechnology Co., Ltd. (Beijing, China). AKT1, α_s1_-casein (CSN1S1) and Cyclin D1 were sourced from Proteintech (Rosemont, IL, USA); P-AKT and p-S6K1 were obtained from Cell Signaling (Danfoss, MA, USA); S6K1, mTOR, P-mTOR, κ-casein (CSN3), β-lactoglobulin (BLG), β-casein (CSN2), prolactin (PRL), prolactin receptor (PRLR), STAT5a, STAT5b and β-actin were obtained from Affinity biosciences, OH, USA; and casein enzymatic hydrolysate was obtained from Beijing Solarbio Science & Technology Co., Ltd. (Beijing, China).

### 2.2. Preparation of Oyster Peptides

According to methods previously described by our group [17], the hydrolysates of the prepared oysters were subjected to ultrafiltration after being dissolved and diluted in distilled water. The supernatant was sequentially fractionated using 10 kDa, 5 kDa, 3 kDa and 1 kDa ultrafiltration membranes to obtain 5–10 kDa, 3–5 kDa, 1–3 kDa and <1 kDa fractions, which were given the names UEC1 to UEC4 in sequence (UEC stands for ultrafiltration components of enzymatic hydrolysates of *Crassostrea hongkongensis*). Next, the filtered solution was freeze-dried and stored at −5 °C until use. The peptide preparation process and the evaluation of cell proliferation are shown in Figure 1.

The cell proliferation rate was determined by the MTT method. The pretreated Sephadex G-15 was loaded into a 1.6 × 100 cm^2^ glass chromatographic column, and a quantity of UEC4 was accurately weighed. The 50 mg/mL solution was prepared with deionized water, and 5 mL of the sample solution was loaded into the column. The sample solution was eluted with deionized water at a flow rate of 0.33 mL/min. The UV detector was used to detect the components at 280 nm, and the components were collected. A sample was collected into a tube every 10 min, and 3.33 mL of the contents of each tube were frozen and dried for further use. A NaCl eluent (1 moL/L) was prepared with a 10 mmoL Tris-buffer solution, 0.50 g of the sample solution was added to 10 mL of the 10 mM Tris-buffer solution to make a final concentration of 50 mg/mL of the sample solution, and the mixture was centrifuged at 12,000× *g* rpm (12,800× g). The upper sample solution was passed through a 0.22 μm filter. The Superdex 30 Increase 3.2/300 pre-packed peptide column was combined with an AKTA Purifier100 protein purification system. The equilibrium solution was continuously washed, and the flow rate was 0.5 mL/min and 0.8 mL/tube. The absorbance of the elution peak was detected at 220 nm. For the RP-HPLC analysis, we took a certain amount of the best proliferation component sample, dissolved it in 1 mL of ultrapure water and prepared a 20 mg/mL concentration of the polypeptide sample solution. The sample solution was injected into a reversed-phase column and subjected to high-performance liquid chromatography. Sample volume: 1 mL; mobile phase: 5–80% (*v*/*v*) methanol containing 0.1% (*v*/*v*) trifluoroacetic acid (TFA); flow rate: 0.8 mL/min; linear elution time, 35 min; column temperature, 35 °C; eluent absorbance was detected at 220 nm.

### 2.3. Peptide Identification by LC-ESI-Orbitrap MS/MS

The samples of UEC4-1 were sent to the Shenzhen Huada Gene Laboratory for sample determination using an Orbitrap FUSION LUMOS mass spectrometer. The offline data were collected using the MaxQuant integrated Andromeda engine. At the spectral level, the filtration was completed with a PSM-level FDR ≤ 1%, and, at the peptide level, the filtration was further carried out with a peptide-level FDR ≤ 1%. The data were processed by searching the Uniprot protein database and the NCBI and Ensembl genome protein databases to obtain the dominant identification results.

### 2.4. Animal Experiment

#### 2.4.1. Animals

Specific-pathogen-free (SPF) healthy SD rats (5–6 weeks old) were purchased from Zhuhai Baishitong Biotechnology Co., Ltd. (Zhuhai, China; animal production license number: SCXK (Guangdong) 2020-0051; certificate number: 44822700004877). There were 90 female rats and 45 male rats. The body weights of the female rats were 150–180 g, and those of the male rats were 180–200 g. The rats were housed separately in standard cages with wood chips and were allowed access to food and water ad libitum in the animal room of the School of Food Science and Technology, Guangdong Ocean University, China (No. IACUC-20190107-02). The system was set to maintain the temperature between 22 and 24 °C and the humidity between 55% and 65%. After 3 weeks of adaptive feeding, sexually mature unfertilized female and male rats at a female:male ratio of 2:1 per cage were mated to breed new pups. After delivery, the number of pups per litter was counted and recorded.

#### 2.4.2. Experimental Design and Treatments

According to methods previously described by other researchers and the dosage determined by our group [17,22,23], bromocriptine mesylate (C_32_H_40_BrN_5_O_5_·CH_4_SO_3_; Da:750.70), a prolactin central inhibitor, was used to establish the postpartum hypogalactia model. Within 24 h of giving birth, 72 female rats and their offspring (12 pups per litter) were randomly divided into nine groups (8 rats in each group): the normal group (NG), the model group (MG), the Buxueshengmi granules (a lactation agent purchased from Jiuzhitang Co., Ltd., Changsha, China) group (BXSMG), the UEC4 high-dose group (UEC4-H), the UEC4 medium-dose group (UEC4-M), the UEC4 low-dose group (UEC4–L), the UEC4-1 high-dose group (UEC4-1-H), the UEC4-1 medium-dose group (UEC4-1-M) and the UEC4-1 low-dose group (UEC4-1-L). From the second day after delivery, the mother rats of each group were gavaged once per day for 7 days. The experimental details are shown in Table 1. The lactating mothers and pups were weighed once a day, and the lactation amount of the mothers and the average body weight gain of the pups during the period were calculated. Any rat deaths were recorded every day.

#### 2.4.3. Net Weights of the Rat Pups and Mammary Gland Parameters

The weight gain per litter was calculated as W_G_ = W_(x)_ − W_(1)_. Here, W_(x)_ is the weight of all the pups in the litter on the 7th day, and W_(1)_ is the weight of all the pups in the litter on the 1st day. On the 8th day, postpartum rats were killed by cervical dislocation, and the pups were weighed. The index weight (I.W.) of each mammary gland was calculated according to Matousek et al. [24]; here, I.W. = organ weight (g)/body weight (g) × 100.

#### 2.4.4. Prolactin (PRL), Prolactin Receptor (PRLR) Levels and Mammary Gland Structure

According to the method of Dong et al. [22], the blood samples were clotted at room temperature and centrifuged at 3000× *g* rpm for 15 min at 4 °C, and the serum was stored at −80 °C for the follow-up assays. Mammary gland tissue (100 mg) samples were rinsed with PBS, homogenized in 1 mL of PBS, stored overnight at −80 °C and then centrifuged at 5000× *g* rpm for 5 min at 4 °C. The supernatant was removed, aliquoted and stored at −80 °C. The collection experiment of the above samples was carried out and completed on 24 August 2021. The serum and tissue homogenates were analyzed using rat prolactin PRL and PRLR ELISA kits (Guangzhou Dingguo Biotechnology Co., Ltd., Guangzhou, China).

The paraformaldehyde-treated mammary gland tissues were dehydrated and embedded in paraffin. Paraffin sections (5 μm) were prepared and stained with hematoxylin and eosin (HE) and observed and photographed under an optical microscope (DMI4000B intelligent inverted fluorescence microscope, Leica, Wetzlar, Germany).

### 2.5. Cell Cultures

The MCF-10A cells were suspended with an appropriate amount of complete medium and transferred into a T25 culture flask. The cells were cultured in a 5% CO_2_ incubator at 37 °C. The culture medium was changed every 48 h. The cells were digested with 0.25% trypsin and passaged at 1:2. After several passages, the cells grew well. According to our previous experimental study, the cells (1 × 10^5^ cells/mL) were seeded in 96-well plates and were administered various concentrations of the sample for 48 h. The final concentrations of UEC1–UEC4 and F1–F4 were 0, 12.5, 25, 50, 100, 200 and 400 μg/mL; those of S1–S4 and P1–P4 were 0, 12.5, 25, 50, 100 and 200 μg/mL. The content of UEC4-1 was 0, 12.5, 25, 50, 100 and 200 μg/mL. A control group of 50 μg/mL casein hydrolysate was set. The MTT method was used to evaluate the proliferation rate of the MCF-10A cells. The proliferation rate was calculated using the following equation:$$\frac{OD_{sample}-OD_{blank}}{OD_{control}-OD_{blank}}\times 100$$

### 2.6. Establishment of the Cell Model for Evaluating the Lactation Effect of UEC4-1

#### 2.6.1. Cell Cycle Synchronization

The cells were starved for 24 h in serum-free medium [25]. MCF-10A cells were inoculated in 6-well plates with a complete medium. When the cell density reached about 50%, the samples were cultured in serum-free DMEM/F12. The normal culture was set as the control, and the cells in the normal culture and starvation culture were collected separately. A cycle analysis was carried out by flow cytometry (FACSCantoTM II, BD Biosciences, San Jose, CA, USA) to detect the synchronization effect.

#### 2.6.2. Cell Cycle Analysis

The experimental groups were as follows: control group, 0 μg/mL; sample groups, 12.5, 25 and 50 μg/mL.

MCF-10A cells with a growth abundance of 80% were subcultured in a 6 cm dish at a rate of 3 × 10^5^/well. After the cells were cultured and had adhered to the wall, the cell culture medium was carefully poured out, and the cells were washed. According to the grouping, the cell culture medium, containing different concentrations of the samples, was added. After 48 h of continuous culture, the cells were washed twice with PBS. After trypsin digestion, the cells were resuspended in PBS. After washing and centrifugation, 70% glacial ethanol was immediately added, and the samples were stored at 4 °C overnight. After the cells were fixed, the cell suspension of the previous day was centrifuged and vortexed to remove the ethanol fixative, and the cells were washed twice with cold PBS. The supernatant was removed, and a propidium iodide (PI) staining solution containing RNase was added. The cells were stained at room temperature in the dark for 15–20 min. The fluorescence intensity in each group was detected by flow cytometry, and the experiment was repeated three times.

#### 2.6.3. Cell Migration Experiment

MCF-10A cells were digested with trypsin. After complete medium was used to inactivate the trypsin, the cells were resuspended and counted. A Transwell chamber (an 8 μm pore-diameter product from Coming) was inoculated with 1 × 10^4^ cells/well. The Transwell chamber was filled with serum-free DMEM/F12 medium. The 24-well plates were filled with a medium containing different concentrations of UEC4-1. After incubation for 16 h, the chamber was taken out. The medium was discarded, and the cells on the inside of the chamber were wiped with cotton swabs. The outer membrane was washed with PBS on both sides and fixed with 4% formaldehyde for 15–20 min. The cells were stained with 0.1% crystal violet and observed under a microscope.

### 2.7. Real-Time PCR Analysis of Gene Expression

Total RNA was extracted from MCF-10A cells using TRIzol (Thermo, Waltham, MA, USA) reagent. A Hifair^®^ II 1st Strand cDNA Synthesis Kit (Takara) was used for reverse transcription to synthesize cDNA. According to the instructions of the kit, RNAiso Plus (Takara, Shiga, Japan) and Hieff^®^ qPCR SYBR Green Master Mix (Low Rox Plus) were used for real-time fluorescence quantitative PCR. Quantitative PCR was performed on ABIViia7 real-time PCR equipment (BIO-RAD T100, Hercules, CA, USA) under the following conditions: denaturation at 95 °C for 10 s, pre-denaturation at 95 °C for 5 s, and annealing at 60 °C for 30 s and 40 cycles. The dissolution curve was collected at 95 °C for 15 s, 60 °C for 60 s and 95 °C for 15 s. The PCR primer series is shown in Table 2. The relative expression levels of the target genes were normalized to β-actin and were calculated by the 2^−^^ΔΔCt^ relative quantitative method. The experiment was repeated three times. The results for different treatments of the same sample were analyzed by t-tests with the SPSS 22.0 statistical analysis software, and an analysis of variance was carried out between multiple sets of data.

### 2.8. Western Blotting

The cell lysate was added into the cell culture dish, the bottom of the culture dish was scraped repeatedly with a cell scraper so that the cells were fully broken and dissolved in the cell lysate, and the cracking process was carried out at a low temperature. The cell lysate was carefully extracted, transferred to a 1.5 mL centrifuge tube and ultrasonicated for 15 s three times. The total protein concentration was detected using the BCA Protein Quantification Kit, and the cell lysate was subjected to boiling water bath denaturation for 10 min. To determine the total protein concentration, the cells were loaded and frozen at −20 °C for further use. The samples were separated by 10% sodium dodecyl sulfate–polyacrylamide gel electrophoresis and transferred to a PVDF membrane at a low temperature. The hybrid membrane was removed, rinsed with TBST for 5 min, blocked with a 5% skim milk powder solution at room temperature for 2 h, rinsed with TBST for 8 min, incubated with the appropriate primary antibody dilution concentration overnight (4 °C), rinsed three times with TBST for 8 min and incubated with a diluted solution of the corresponding secondary antibody at 37 °C for 2 h. Finally, the hybrid membrane was developed by chemical fluorescence imaging. The IPP software version 6.0 was used for the quantitative analysis.

### 2.9. Statistical Analysis

All the quantitative data are presented as the means ± standard deviations (mean ± S.D). The data were analyzed using SPSS version 22.0. The experimental values were evaluated by a one-way ANOVA following the LSD method for multiple comparisons among the groups; *p*-values below 0.05 and 0.01 were considered significant and very significant, respectively.

## 3. Results

### 3.1. Preparation of Lactation Activation Peptides from Oyster Hydrolysate

In our study, the hydrolysates of *C. hongkongensis* were used as raw materials to obtain four components (UEC1–UEC4) by ultrafiltration, and the best component for the proliferation of MCF-10A cells, UEC4, was determined by the MTT method (Figure 2A). Four main fractions of UEC4, F_1_–F_4_, were isolated by Sephadex G-15 gel chromatography (Figure 3A), and the best fraction, F_3_, was determined by an evaluation of cell proliferation (Figure 2B). Four main fractions, S_1_–S_4_ (Figure 3B), were isolated and purified by chromatography on a Superdex prepacked column combined with an AKTA protein purification system, and the best fraction, S_3_, was evaluated for its proliferation effects (Figure 2C). Four fractions, P_1_–P_4_ (Figure 3C), were prepared by the separation and purification of the S_3_ fraction by RP-HPLC, and the best fraction, P_3_ (UEC4-1) (Figure 2D), was obtained.

### 3.2. Peptide Identification by LC-ESI-Orbitrap MS/MS

As shown in Figure 4 and Table 3, the results indicated that the components of UEC4-1 were unmodified peptides with sequences of seven amino acids: Val–Gly–Arg–Thr–Asn–Ser–His (VGRTNSH), Val–Gly–Thr–Glu–His–Arg–Lys (VGTEHRK), Asn–His–Ile–Ser–Trp–Ala–Ala (NHISWAA), Ser–Tyr–Lys–Cys–Arg–Asn–Ser (SYKCRNS), Phe–Pro–Val–Ala–Leu–Gly (FPVALMG) and Val–Gly–Met–Ile–Gly–Phe–Leu (VGMIGFL). According to various factors, the UEC4-1 component was determined to be a polypeptide with VGRTNSH as the main component (70.72%).

### 3.3. Effect of UEC4 and UEC4-1 Intake on the Lactation Function of Rats

#### 3.3.1. Body Weight Gain of Offspring and Breast Organ Index of Female Rats

Figure 5A,B show the growth of the rat pups in each group after 7 days of treatment. The rat pups in the normal group grew well after 7 days of gavage, and the mother rats were normal. The body weight gain of the model group was significantly lower than that of the normal group, and the difference was extremely significant (*p* < 0.01). There were no significant differences among the BXSMG, UEC4-H, UEC4-1-H and normal groups (*p* > 0.05). The body weight gain of the BXSMG, UEC4-H and UEC4-1-H groups was significantly higher than that of the model group (*p* < 0.01). The results show that the BXSMG, UEC4-H and UEC4-1-H treatments could significantly increase the body weight gain of each litter, indicating that the lactation ability of the postpartum female rats in the BXSMG, UEC4-H and UEC4-1-H groups was close to that of normal female rats.

The data analysis results for the breast organ index of the maternal rats are shown in Figure 5C. There was a significant difference between the model group and the normal group (*p* < 0.01). There were no significant differences among the BXSMG group, the UEC4-1-H group and the normal group (*p* > 0.05). The breast index of the BXSMG, UEC4-H, UEC4-M, UEC4-L, UEC4-1-H, UEC4-1-M and UEC4-1-L groups increased by 35.71%, 28.57%, 21.42%, 14.29%, 32.14%, 25.00% and 17.86%, respectively.

Our results show that postpartum lactating rats could be fed normally by the administration of Buxueshengmi granules (BXSMG), the <1 KDa ultrafiltration fraction of the oyster hydrolysate (UEC4) and the purified polypeptide (UEC4-1).

#### 3.3.2. Serum PRL Levels

The results for the maternal serum PRL levels are shown in Figure 5D. The serum PRL content of the female rats in the model group was lower than that in the normal group, and the difference was extremely significant (*p*
*<* 0.01). There were no significant differences in the serum PRL content among the BXSMG group, the UEC4-1 high-dose groups and the normal group. Compared with those in the model group, the serum PRL levels of the BXSMG, UEC4-H, UEC4-1-H and UEC4-1-M groups increased by 68.84%, 49.44%, 61.63% and 49.21%, respectively, and the differences were extremely significant (*p* < 0.01). The serum PRL levels in the UEC4-M group increased by 48.98% (*p* < 0.05). The results show that Buxueshengmi granules, high and medium doses of purified peptides (UEC4-1) and the <1 KDa ultrafiltration of enzymatic hydrolysate of oyster (UEC4) could significantly increase the level of serum PRL in breast-deficient maternal rats, and the medium dose of UEC4 could significantly increase the level of serum PRL in milk-deficient maternal rats, but the effect in the low-dose group was not significant.

#### 3.3.3. Tissue PRL and PRLR Levels

As shown in Figure 5E, the content of PRL in the mammary glands of the model group was significantly lower than that of the normal group (*p* < 0.01). There were no significant differences in mammary gland PRL content among the BXSMG, UEC4-H and high-, medium- and low-dose groups of UEC4-1 compared with the normal group. Compared with the model group, the PRL in the BXSMG group increased by 11.17%, and the PRL in the UEC4-H and UEC4-1-H groups increased by 7.49% and 8.13%, respectively. The results show that Buxueshengmi granules could significantly increase the level of PRL in the breast tissue of breast-deficient maternal rats.

As shown in Figure 5F, the PRLR content of mammary glands in the model group was significantly lower than that in the normal group, while there were no significant differences in breast PRLR between the BXSMG, UEC4-H and UEC4-1-H groups and the normal group. Compared with the model group, the PRLR in the BXSMG and UEC4-1-H groups increased by 25.48% and 28.33%, respectively, and the PRLR in the UEC4-H group and the UEC4-1-M group increased by 23.13% and 18.08%, respectively. The results show that UEC4-1Mel H and Buxueshengmi granules could significantly increase the level of PRLR in the breast tissue of breast-deficient maternal rats. The high-dose UEC4-1 group showed the best effects.

#### 3.3.4. Mammary Gland Structure

The pathological structure of the mammary glands of the maternal rats is shown in Figure 6. The breast acini of the normal group were numerous and dense, the acini were neatly arranged and complete, the diameter of the acini was larger, the interacinar space was compact, the mammary lobule was clear and the duct was thicker. Compared with the normal group, the adipose connective tissue of the model group was significantly increased, lobulation was not obvious, the shape of the acini was irregular, the acinar tissue was rare and scattered, the acini were narrow, only a few secretions could be seen and the structure showed obvious atrophy. Compared with the model group, the breast acini of the BXSMG group increased significantly and were arranged neatly, the diameter of the gland duct was larger and the mammary lobule basically returned to normal. The morphology of the mammary gland in the UEC4-H group was similar to that of the BXSMG group. The mammary acinar ducts of the UEC4-1 high-dose group were significantly dilated. The results show that Buxueshengmi granules, UEC4-1 and UEC4 (high- and medium-dose groups) could partially develop breast tissue to the lactation stage and increase the number of glandular epithelial cells in postpartum milk-deficient rats. In the UEC4-1 high-dose group, the number of breast acini increased significantly, the diameter of the mammary duct dilated obviously, the interlobular fat connective tissue decreased, and the mammary lobule basically returned to normal. The intervention effect in the medium-dose UEC4-H group was slightly lower than that in the BXSMG group, and the effect in the low-dose group was not obvious.

### 3.4. Evaluation of Lactation Effect of UEC4-1

#### 3.4.1. Synchronization of MCF-10A Cells by the Serum Starvation Assay

Figure 7A shows the percentages of cells in the three repeated experiments. The proportions of the G0/G1, S and G2/M phases of MCF-10A cells in the normal culture were 52.82%, 35.24% and 11.94%, respectively. After 48 h of starvation in a serum-free medium, almost all the cells were synchronized in the G0/G1 and S phases, and the proportions were 96.04%, 3.76% and 0.20%, respectively. The results show that, after 48 h of starvation in serum, MCF-10A cells could achieve the G0/G1 phase.

#### 3.4.2. UEC4-1 Promotes the G1–S Phase Transition Process of MCF-10A Cells

The results in Figure 7B show that UEC4-1 could significantly promote the proliferation of MCF-10A cells. The cells cultured in serum-free medium were used as the control group, and the sample group was treated with 50 μg/mL UEC4-1. Cells cultured for 0–24 h were collected for periodic detection. The results show that the percentage of the S phase in the control group was maintained at about 6%, and the UEC4-1 treatment group began to enter the S phase at 12 h, while the proportion of the S phase at 18 h reached about 60% and then gradually returned to normal with the passage of time. This indicated that cell cycle progression was induced by the addition of UEC4-1, and UEC4-1 could significantly promote the transition of MCF-10A cells from the G1 phase to the S phase.

#### 3.4.3. UEC4-1 Promotes Migration

As shown in Figure 7C, the effect of UEC4-1 on the migration ability was tested. UEC4-1 treatment significantly increased the migration ability of MCF-10 A cells; 20 visual fields were randomly selected for counting, and the average number of cells in the UEC4-1 treatment group was significantly different from that in the control group (*p* < 0.01).

#### 3.4.4. Effect of UEC4-1 on the Proliferation of MCF-10A Cells

The results are shown in Figure 7D. The differences between the 12.5, 25 and 50 μg/mL treatment groups and the blank group were extremely significant (*p* < 0.01). Compared with the enzymatic hydrolysis casein group, under the treatment with 50 μg/mL, the proliferation rate was 174.58% (*p* < 0.01). Under the treatment with 100 μg/mL, the proliferation rate showed a downward trend.

### 3.5. Effects of UEC4-1 on PRL and PRLR Expression

The results of the data analysis are shown in Figure 8. Compared with the control group, the relative expression of the mRNA of PRL and PRLR increased in a dose-dependent manner (12.5–50 µg/mL of UEC4-1). Under the treatment with 50 μg/mL, the relative expression of PRL and the mRNA of PRLR increased by 6.13 and 2.17 times, respectively. The results show that 50 μg/mL of UEC4-1 could significantly upregulate the gene expression of prolactin PRL and its receptor, PRLR, in MCF-10A cells (Figure 8A). The changes in protein expression were detected by Western blot analysis. The results show that, compared with the control group, 25 μg/mL UEC4-1 could significantly increase the protein expression of PRL and PRLR, and 50 μg/mL UEC4-1 could significantly increase the protein expression of PRL and PRLR (*p* < 0.01) (Figure 8B).

### 3.6. Effects of UEC4-1 on the Expression of Genes Related to Milk-Protein Synthesis

It can be seen in Figure 9A that, compared with the control group, 25 μg/mL UEC4-1 significantly increased the expression levels of casein (CSN1S1, CSN2 and CSN3) genes (*p* < 0.05), and 50 μg/mL UEC4-1 increased the total relative expression levels of these three casein mRNAs in MCF-10A cells by 49.34-, 44.96- and 62.49-fold, respectively, which are extremely significant differences (*p* < 0.01). Figure 9A shows that, compared with the control group, the 50 μg/mL UEC4-1 group showed significantly increased gene expression levels of cyclin CCND1 (*p* < 0.05), by 2.03-fold.

As can be seen from Figure 10A, compared with the control group, the protein expression of casein β-lactoglobulin, α_S1_-casein, β-casein and κ-casein increased in a dose-dependent manner (12.5–50 µg/mL). In the 50 μg/mL group, the expression of the BLG protein increased significantly (*p* < 0.05). The protein expression of CSN1S1 in the 25 μg/mL group increased significantly (*p* < 0.05), and it also increased significantly in the 50 μg/mL group *(p* < 0.01). The protein expression of CSN2 in the 12.5 μg/mL group increased significantly (*p* < 0.05), and it also increased significantly in the 25 and 50 μg/mL groups (*p* < 0.01). The expression of the CSN3 protein in the 25 μg/mL group increased significantly (*p* < 0.05), and it also increased significantly in the 50 μg/mL group (*p* < 0.01).

The results show that, compared with the control group, the expression of the CCND1 protein in the 25 μg/mL group and 50 μg/mL group increased significantly (*p* < 0.01), as shown in Figure 10A.

Our results show that 50 μg/mL UEC4-1 could significantly increase the gene expression levels of three caseins (CSN1S1, CSN2 and CSN3) in MCF-10A cells. With an increase in the UEC4-1 concentration, the protein expression of CSN1S1, CSN2, CSN3 and CCND1 increased significantly compared with the control group. Different concentrations of UEC4-1 could positively regulate the synthesis of major milk proteins and cyclins in MCF-10A cells.

### 3.7. Effects of UEC4-1 on Key Genes of the Lactation Signaling Pathway

Figure 9B shows that the relative mRNA expression of mTOR, AKT1, S6 KB1, STAT5 A and STAT5 B increased in a dose-dependent manner (12.5–50 µg/mL) compared with the control group. The expression levels reached their highest values at 50 µg/mL, increasing by 1.63-, 1.64-, 1.47-, 1.97- and 1.93-fold, respectively, and the differences were significant (*p* < 0.05). The 25 µg/mL group showed significantly increased protein expression of mTOR and AKT1 (*p* < 0.05). The 50 µg/mL group showed significantly increased protein expression levels of mTOR and AKT1 (*p* < 0.01), whereas the 25 μg/mL UEC4-1 group showed significantly increased protein expression levels of P-mTOR, P-AKT and P-S6KB1 (*p* < 0.05). The 50 µg/mL UEC4-1 group showed significantly increased protein expression levels of P-mTOR, P-AKT and P-S6KB1 (*p* < 0.01) in Figure 10B.

## 4. Discussion

Recent studies have shown that small peptides play an important role in the mechanism of milk-protein synthesis. The concentration and amino acid composition of small peptides are also important factors affecting milk-protein synthesis, while little is known about the underlying mechanisms. The present study demonstrated that the cell proliferation of MCF-10A cells treated with the ultrafiltration component UEC4 was significantly increased after the fractionation purification of this component. Then, the UEC4-1 fraction significantly promoted the proliferation of MCF-10A cells. These results showed that UEC4-1 promoted the proliferation of MCF-10A cells in a concentration-dependent manner. Concentrations lower than 50 μg/mL could not meet the needs of cell growth, while concentrations higher than at 50 μg/mL inhibited cell growth. Similarly, some studies have shown that dipeptides and tripeptides (Met–Met and Thr–Phe–Phe) have higher absorptive efficiency than free amino acids, but only appropriate concentrations of small peptides can promote the growth, proliferation and milk-protein secretion in BMECs [20,21,26,27]. For example, the addition of Met–Met at 80 μg/mL best promoted the synthesis of casein in BMECs, and too low or too high a concentration may inhibit growth and affect casein’s gene expression [21].

In order to study the effect of UEC4-1 on lactation and its regulatory mechanism, a model for evaluating the effect of polypeptides on lactation was established using the normal line of MCF-10A HMEC cells. Cell proliferation, migration and the cell cycle process were reported [25]. The results from the present study show that UEC4-1 significantly promoted the proliferation of MCF-10A cells, cell transition from the G1 phase to the S phase and the migration of cells. The successful establishment of this model laid a foundation for the next step of screening the regulatory molecules of the polypeptide lactation-related signal pathway. Furthermore, our data elucidated that, compared with the control, 50 μg/mL UEC4-1 could significantly increase the gene expression of cyclin D1 in MCF-10A cells, promote the proliferation of MCF-10A cells and stimulate the synthesis of lactoprotein in HMECs. Similarly, Wang et al. found that the addition of 80 μg/mL Met–Met promoted the transition of BMEC cells from the G1 phase to the S phase, increased the expression of cyclin D1, promoted the proliferation of BMECs and stimulated the synthesis of lactoprotein in MECs [21].

The data from the present study indicate that, with an increase in the UEC4-1 concentration, 50 µg/mL of UEC4-1 significantly increased the mRNA expression of three caseins, CSN2, CSN1S1 and CSN3. The protein expressions of CSN1S1, CSN2 and CSN3 were significantly higher than those in the control group. Moreover, UEC4-1 positively regulated the synthesis of milk protein in HMECs. The treatment of HMECs with 50 µg/mL UEC4-1 significantly increased the mRNA abundance of the signaling pathway genes mTOR, AKT and STAT5. At the level of phosphorylation, mTOR, AKT and S6KB1 were significantly increased when compared with the cells without UEC4-1 (the control group). Therefore, we speculate that UEC4-1 may increase casein mRNA expression by activating mTOR and phosphorylating its downstream molecule S6KB1, thereby regulating the expression and phosphorylation of mTOR pathway proteins. Cai et al. studied the effect of bioactive peptide fractions (OPH3-1) on β-casein expression in mouse mammary epithelial cells (HC11). OPH3-1 significantly stimulated cell proliferation and β-casein synthesis in HC11 cells and enhanced the mRNA abundance of the JAK2–STAT5 and mTOR signaling molecules. These data were consistent with β-casein upregulation [18]. Yang et al. showed that Met–Met promoted α_s1_-casein synthesis in cultured bovine mammary gland explants, and this stimulation may have been mediated by enhanced intracellular substrate availability and by the activation of the JAK2–STAT5 and mTOR signaling pathways [20]. At the translation level, mTOR regulated cell growth and protein synthesis by binding mRNA and phosphorylating the eukaryotic initiation factor Eif4b-binding protein ribosomal 6S kinase S6K1 [28,29]. In addition, Wang et al. elucidated the properties of the peptide transporter and its effects on β-casein synthesis in BMECs, which included Met–Met being taken up by PepT2 and enhancing cell proliferation and β-casein synthesis by activating the JAK2–STAT5 and mTOR signaling pathways in BMECs [21].

Prolactin (PRL) is a polypeptide hormone synthesized in and released from the anterior pituitary gland. It is known that PRL plays an essential role in milk synthesis and secretion during lactation. which is needed for the onset and maintenance of lactation. Moreover, the PRL concentration was also positively correlated with the mammary gland wet weight in gilts and the milk yield of sows [30]. Our in vivo data showed, at high-dose (0.6 g/kg of UEC4-1), significantly increased PRL concentration in the serum as well as significantly increased PRL and PRLR concentration in breast tissue. These results indicate that UEC4-1-H could effectively increase the lactation yield and breast development in lactating rats induced by bromocriptine. The effect was better than that in the Buxueshengmi granule group. 

To explore the mechanism of lactation regulation, the in vitro data from the present study demonstrated that 50 μg/mL of UEC4-1 significantly increased the mRNA abundance of PRL and PRLR compared with the control. The possible mechanism may be through the activation of AKT and STAT5, promoting the proliferation of HMECs and stimulating the expression of PRLR. A study reported that pituitary PRL binding to PRLR could activate JAK2–STAT5 signaling, which is required for lobuloalveolar development in the mammary gland during pregnancy [31]. Lin et al. showed that quercetin may promote the proliferation of primary MECs and stimulate the expression of PRLR through AKT activation in vitro [32]. Another study found that the major pathway mediating prolactin action is the Janus kinase 2 (JAK2)/signal transducer and activator of transcription (STAT) pathway, which is activated by the long form of the prolactin receptor (PRLR). In addition to the JAK/STAT pathway, under some conditions, PRL may activate the signal transduction cascade (phosphatidylinositol-4,5-bisphosphate 3-kinase [PI3K]) of the phosphoinositol-3-kinase/protein kinase B (or Akt) pathway as well as the mitogen-activated protein kinase or extracellular regulated kinase pathways [33].

In summary, the results indicate that the components of UEC4-1 were unmodified peptides with amino acid sequences of seven amino acids each: Val–Gly–Arg–Thr–Asn–Ser–His (VGRTNSH), Val–Gly–Thr–Glu–His–Arg–Lys (VGTEHRK), Asn–His–Ile–Ser–Trp–Ala–Ala (NHISWAA), Ser–Tyr–Lys–Cys–Arg–Asn–Ser (SYKCRNS), Phe–Pro–Val–Ala–Leu–Gly (FPVALMG) and Val–Gly–Met–Ile–Gly–Phe–Leu (VGMIGFL). According to various factors, the UEC4-1 component was determined to be a polypeptide with VGRTNSH as the main component (70.72%). Yang et al. and Zhou et al. have shown that dipeptides and tripeptides containing restricted amino acids such as methionine (Met), lysine (Lys), threonine (Thr) and phenylalanine (Phe) can promote the synthesis of milk protein in dairy cows MECs [20,27]. Wu et al. showed that Leu, Ile and Val could activate the mTOR signaling pathway, enhancing milk-protein synthesis [8]. The polypeptides of UEC42-1 have the structural characteristics of amino acids that promote lactation.

Interest in the role of peptides in milk-protein synthesis is increasing. Although the role of peptides in milk-protein synthesis is clearly established, little is known about the underlying mechanisms. Therefore, it was necessary to study the lactation mechanism for oyster peptides in this study.

## 5. Conclusions

Postpartum hypogalactia is a problem that has long plagued nursing mothers. Therefore, the prevention and treatment of postpartum hypogalactia is of vital significance for lactating mothers and infants. In conclusion, our data indicate that UEC4-1 may be related to the activation of AKT and STAT5, stimulating the expression of the prolactin receptor, upregulating cyclin D1’s mRNA abundance and promoting the transition of the cell cycle from the G1 phase to S phase. UEC4-1 could increase casein synthesis in HMECs by increasing cell proliferation and activating the mTOR signaling pathway. Our results suggest that the increase in milk-protein synthesis produced by UEC4-1 may be the result of the joint action of the PRL/AKT/STAT5 and mTOR/S6KB1 signaling pathways. The possible mechanism is shown in Figure 11. However, the structure of the UEC4-1 peptide needs to be further clarified. The regulatory mechanism of the peptides on casein synthesis and the PRL/AKT/STAT5 and mTOR/S6KB1 signaling pathways need to be studied in vivo.

## Figures and Tables

**Figure 1 nutrients-14-01786-f001:**
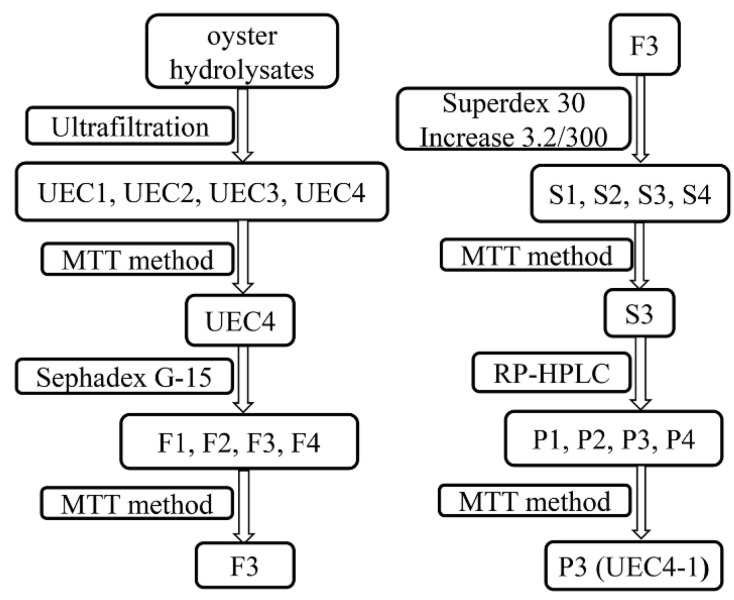
Preparation and lactation activity of polypeptides.

**Figure 2 nutrients-14-01786-f002:**
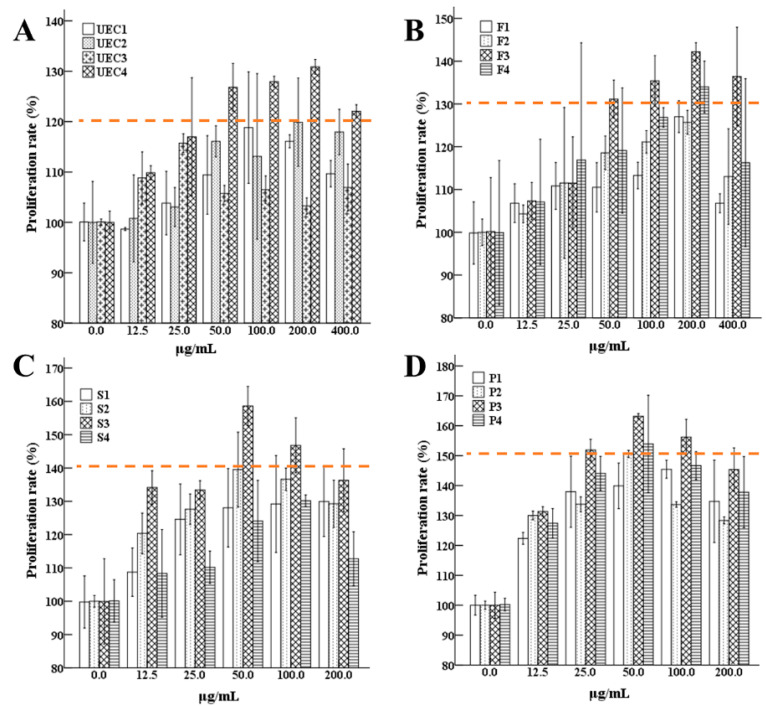
The proliferative effects of the isolated and purified components were evaluated. (**A**): Ultrafiltration fractions UEC1–UEC4; (**B**): fractions F1–F4; (**C**): fractions S1–S4; (**D**): fractions P1–P4.

**Figure 3 nutrients-14-01786-f003:**
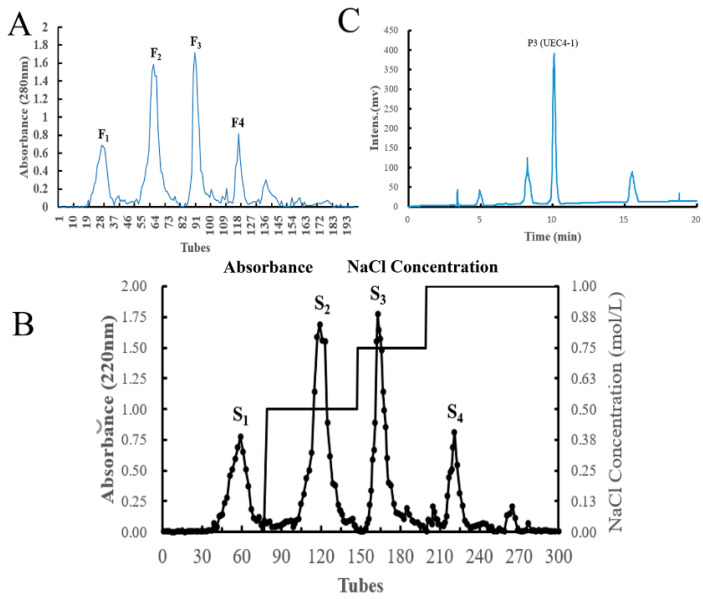
Separation and purification map of the fractions of enzymatic hydrolysis products from oyster. (**A**): Sephadex G-15 separation; (**B**): Superdex pre-packed column separation; (**C**): RP-HPLC analysis.

**Figure 4 nutrients-14-01786-f004:**
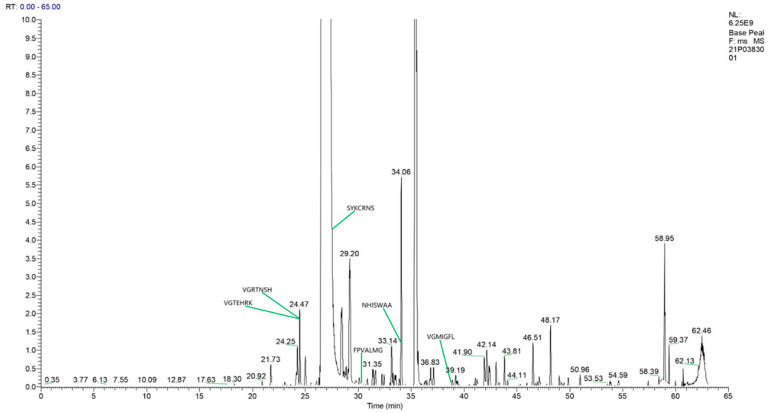
Total ion chromatogram of peptides from oyster hydrolysates (UEC4-1).

**Figure 5 nutrients-14-01786-f005:**
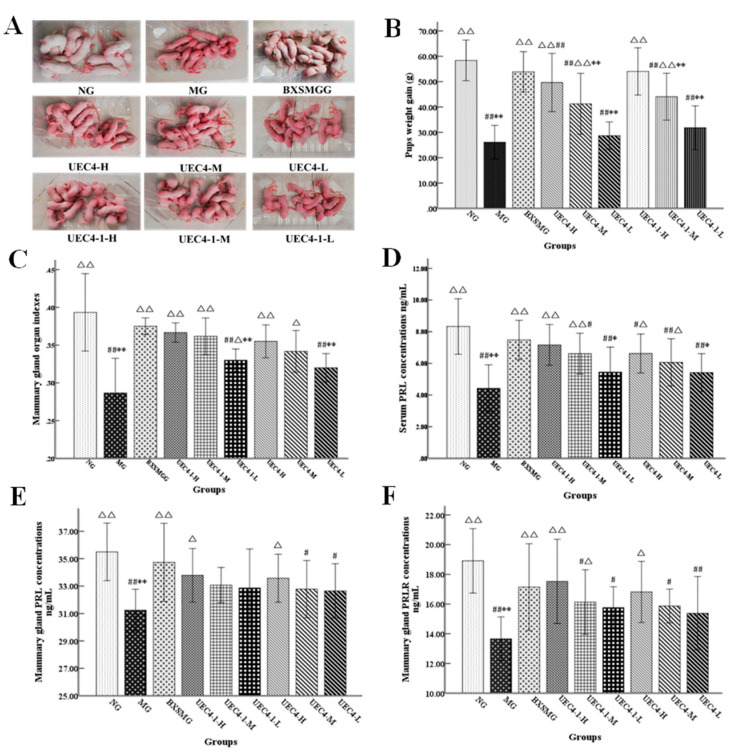
Effects of UEC4 and UEC4-1 on the indices of rats. (**A**) The growth of rats in each group; (**B**) Comparison of body weight gain of offspring rats; (**C**) Mammary gland organ indexes of female rats; (**D**) Serum PRL concentrations of female rats; (**E**) Mammary gland PRL concentrations of female rats; (**F**): Mammary gland PRLR concentrations of female rats. # significantly different from the normal group at *p* < 0.05, ## significantly different from the normal group at *p* < 0.01; Δ significantly different from the model group at *p* < 0.05, ΔΔ significantly different from the model group at *p* < 0.01; * significantly different from the BXSM group at *p* < 0.05, ** significantly different from the BXSM group at *p* < 0.01.

**Figure 6 nutrients-14-01786-f006:**
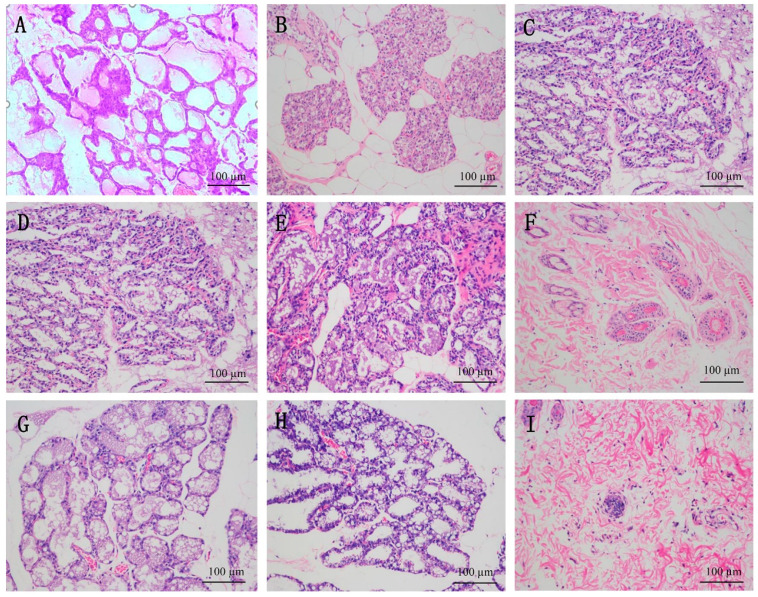
Mammary gland tissue in different groups of female rats under H&E staining; scale bars: 100 μm. (**A**) Normal group (NG); (**B**) model group (MG); (**C**) Buxueshengmi granules group (BXSMG); (**D**) UEC4 high-dose group (UEC4-H); (**E**) UEC4 medium-dose group (UEC4-M); (**F**) UEC4 low-dose group (UEC4-L); (**G**) UEC4-1 high-dose group (UEC4-1-H); (**H**) UEC4-1 medium-dose group (UEC4-1-M); (**I**) UEC4-1 low-dose group (UEC4-1-L).

**Figure 7 nutrients-14-01786-f007:**
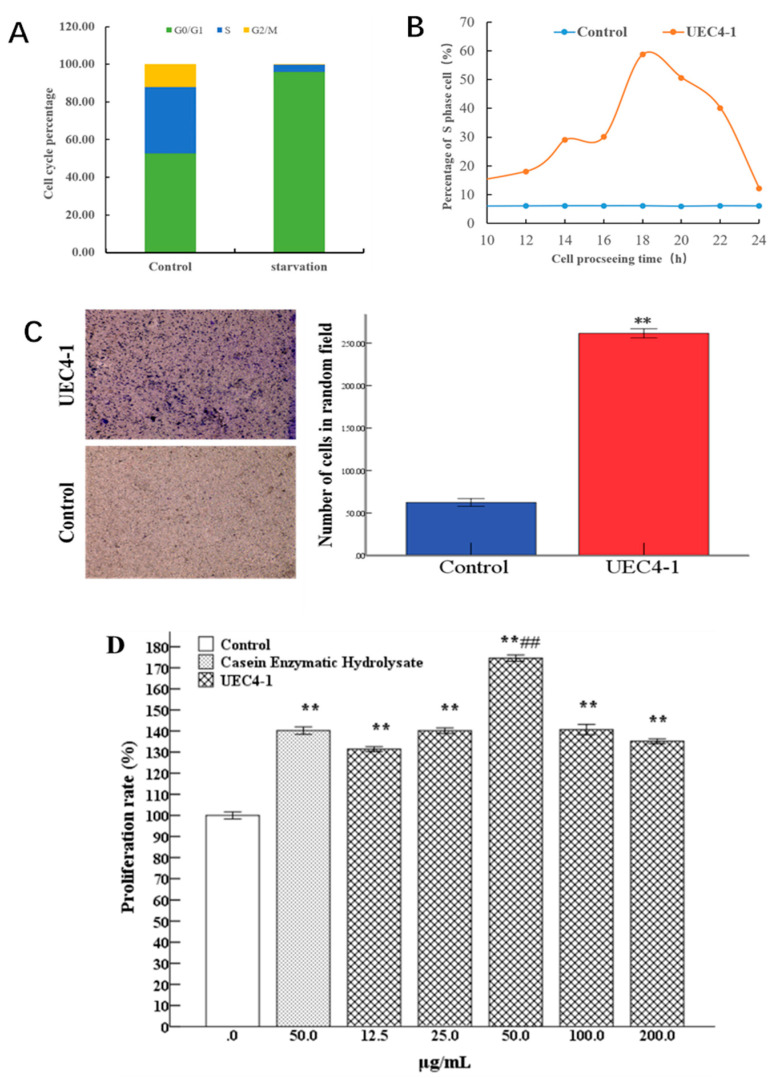
Evaluation of the effects of UEC4–1 on lactation. (**A**) Cell synchronization results; (**B**) UEC4-1 promoted the G1-S phase process of MCF-10A cells; (**C**) UEC4-1 promoted MCF 10A cells. ** significantly different from the control group at *p* < 0.01; (**D**) Effects of UEC4-1 on proliferation of MCF-10A cells. ** significantly different from the control group at *p* < 0.01; ## significantly different from the casein enzymatic hydrolysate group at *p* < 0.01.

**Figure 8 nutrients-14-01786-f008:**
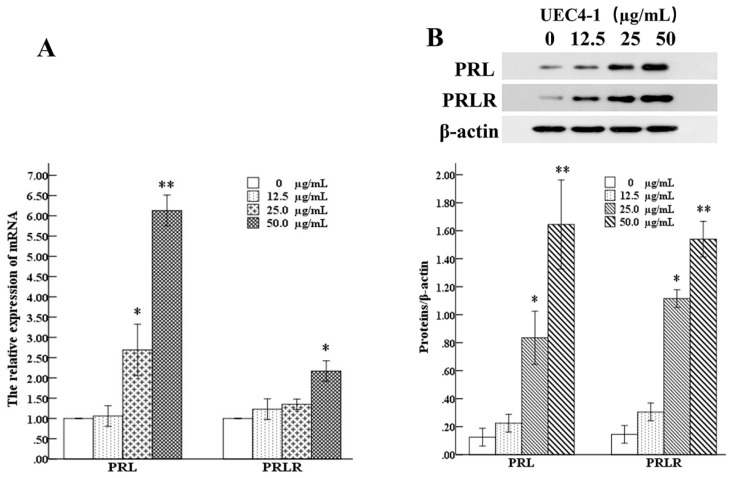
Effects of UEC4-1 on PRL and PRLR expression. (**A**) The expression of PRL and PRLR in mammary gland; (**B**) The protein levels of PRL and PRLR in mammary gland. * significantly different from the control group at *p* < 0.05, ** significantly different from the control group at *p* < 0.01.

**Figure 9 nutrients-14-01786-f009:**
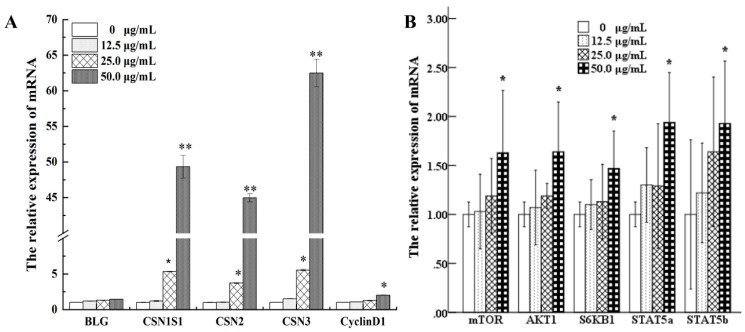
Effect of UEC4-1 on the expression of genes related to milk-protein synthesis. (**A**) The expressions of BLG, CSN1S1, CSN2, CSN3 and CyclinD1 in MCF-10A cells were detected using qRT-PCR; (**B**) Changes in the level of lactation- and signaling-related gene expression were detected using qRT-PCR. * significantly different from the control group at *p* < 0.05, ** significantly different from the control group at *p* < 0.01.

**Figure 10 nutrients-14-01786-f010:**
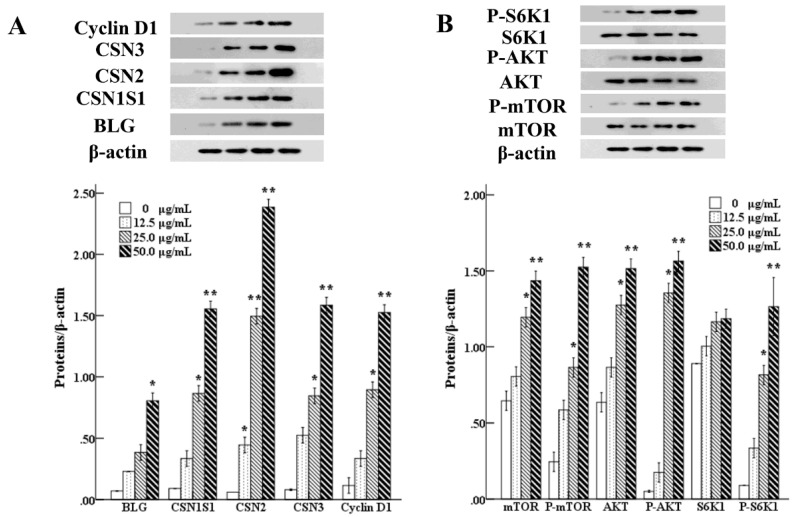
Effects of UEC4-1 on key genes of the lactation signaling pathway in MCF-10A cells. (**A**) Level of BLG, CSN1S1, CSN2, CSN3 and CyclinD1 protein were assessed by Western blotting analysis; (**B**) Changes in the level of lactation- and signaling-related protein were assessed by Western blotting analysis. β-Actin was assessed as a loading control. * significantly different from the control group at *p* < 0.05, ** significantly different from the control group at *p* < 0.01.

**Figure 11 nutrients-14-01786-f011:**
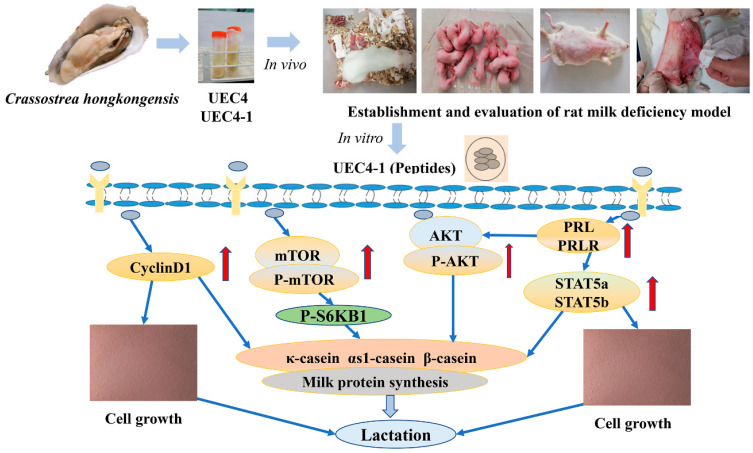
Possible mechanism of UEC4-1.

**Table 1 nutrients-14-01786-t001:** Grouping and treatment of female rats.

Group	Rats/Pups	Weight (g)	Dosage (g/kg)	Intragastric Administration Method
NG	8/12	288.85 ± 1.20	-	Distilled water, 1 mL/100 g
MG	8/12	287.85 ± 1.33	0.005	Bromocriptine, distilled water, both 1 mL/100 g
BXSMG	8/12	290.20 ± 1.84	2.66	BXSM, bromocriptine, distilled water, all 1 mL/100 g
UEC4-H	8/12	291.83 ± 1.65	1.2	UEC4, bromocriptine, distilled water, all 1 mL/100 g
UEC4-M	8/12	287.83 ± 1.69	0.6	UEC4, bromocriptine, distilled water, all 1 mL/100 g
UEC4-L	8/12	289.20 ± 1.19	0.3	UEC4, bromocriptine, distilled water, all 1 mL/100 g
UEC4-1-H	8/12	290.58 ± 1.98	0.6	UEC4-1, bromocriptine, distilled water, all 1 mL/100 g
UEC4-1-M	8/12	288.42 ± 1.52	0.3	UEC4-1, bromocriptine, distilled water, all 1 mL/100 g
UEC4-1-L	8/12	287.92 ± 1.95	0.15	UEC4-1, bromocriptine, distilled water, all 1 mL/100 g

**Table 2 nutrients-14-01786-t002:** Primer sequences for qRT-PCR analysis.

Gene	Sequences (5′–3′)	Length (bp)	Accession (Gene ID)
PRLR	F: GTGGATCTCTGTGGCTGTCC	184	5618
	R: GCATCCCAAGGCACTCAGTA		
PRL	F: CTGGTGTCAAACCTGCTCCT	127	5617
	R: GGATGTAGTGGGACAGGACG		
AKT1	F: CAGGATGTGGACCAACGTGA	137	207
	R: AAGGTGCGTTCGATGACAGT		
BLG	F: ACTATACGGTGGCGAACGAG	101	5047
	R: CATCATGCTCTGGATGGGGG		
CSN1S1	F: AGGGCACCTAATCAGAGGGT	101	1446
	R: AATTGATGGCACTTACAGAACTGG		
CSN2	F: GCAGGTCCCTCAGCCTATTC	120	1447
	R: ACAGCTCTCTGAGGGTAGGG		
CSN3	F: AAATAGCCACCCACCCACTG	141	1448
	R: GCAGGAGCTGGTGTAGGTTC		
Cyclin D1	F: GATGCCAACCTCCTCAACGA	163	595
	R: ACTTCTGTTCCTCGCAGACC		
S6K1	F: TTATTGGCAGCCCACGAACA	184	6198
	R: CGTATTGGAAGTGGTGCCGA		
mTOR	F: TGCTGAACTGGAGGCTGATG	122	2475
	R: TGGCTCTCCAAGTTCCACAC		
STAT5A	F: TACCCACAGAACCCTGACCA	103	6776
	R: TTGGTCGGCGTAAGAGTTCC		
STAT5B	F: GACCAAGTTTGCAGCCACTG	136	6777
	R: ATTGCGGGTGTTCTCGTTCT		
β-actin	F: AGACCTGTACGCCAACACAG	132	60
	R: CGCTCAGGAGGAGCAATGAT		

**Table 3 nutrients-14-01786-t003:** Main peptide sequences of UEC4-1.

Sequence	Peptide Sequence	Molecular Mass (Da)	*m*/*z*	Scores	Intensity (%)
1	VGRTNSH	769.38	384.69	21.55	70.72
2	VGTEHRK	825.45	412.72	26.11	4.57
3	NHISWAA	735.39	367.69	16.15	4.96
4	SYKCRNS	913.41	456.70	14.78	1.02
5	FPVALMG	733.38	366.69	3.62	1.99
6	VGMIGFL	735.39	367.69	1.62	4.59

## Data Availability

The data presented in this study are available from the corresponding author upon reasonable request.

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
