# Peer review of "Lactation Activity and Mechanism of Milk-Protein Synthesis by Peptides from Oyster Hydrolysates"

_nutrients, 2022, doi:10.3390/nu14091786_

Round 1

Reviewer 1 Report

The paper "Lactation Activity and Mechanism of Milk-Protein Synthesis from the Peptides of Oyster Hydrolysates" is an original article that that demonstrates the effect of a specific peptide fraction (UEC4-1) isolated from oyster meat on lactation activity in rats. Moreover, this research seeks to elucidate the mechanism by which the UEC4-1 fraction can influence lactation in rats.

Section 1. Introduction.

The language requires English editing due to sentences hard to read or syntactic errors. The title should be also reworded. Some expressions need to be replaced with more appropriate ones: e.g., "poor breast milk production" can be replaced with low breast milk supply, "Val could increase casein’s mRNA abundance" can be replaced with Val could increases transcription of casein's genes.

Define abbreviations when they are first used in the text (e.g. - AAs, EAAs).

For Latin words (e.g. in vitro) it is recommended to put it in italics.

In some sentences the conjunction "and" is used too often: e.g.," the quality of dairy products, and milk yield and quality".

The phrase "In our previous study, an enzymatic hydrolysate of
Crassostrea hongkongensis could obviously increase hourly lactation in overloaded lactating rats and pup weight, significantly promoted the expansion and filling of the mammary gland’s acinar cavity in these rats and promoted lactation" needs to be reworded. Idem for phrases In this study, we studied the potential efficacy and mechanism of the active peptide UEC4-1 for improving postpartum hypogalactia in rats with hypogalactia. The effect of
UEC4-1 on the proliferation of HMECs was studied to lay the foundation for structural analysis and research into the lactation mechanism.

Section 2. Materials and methods.

The legend of figure 1 is too loaded. The description of the method by which the fractions were obtained can be given as a separate paragraph on materials and methods.

The quality of all the images used in the article is very poor. Try replacing them with better images.

ad libitum - in italics

In the section 2.4.2. the authors mention From the second day after delivery, each group was gavaged once per day for 7 days. It should be mentioned more clearly that only mothers were fed by gavage. From the text above it appears that each group is made up of mothers and their puppies.

In the same section is mentioned Any rat deaths were recorded every day. The authors need to be more specific: how many animals died in each group? This can influence the results.

According to the section 2.4.3.,  the weight gain per litter was calculated as the weight of all the pups in the litter on the 7th day and the weight of all the pups in the litter on the 1st day. Has it been taken into account that some animals have died?

Related to the collection of blood samples (section 2.4.4) to determine the concentration of prolactin and prolactin receptor, please mention in the text when exactly they were collected.

The following sentence is incomplete (section 2.6.1): The serum starvation method was used....please, rephrase it. In the same section, mention the type of the apparatus that was used to detect the synchronization effect.

Section 2.62. Define good growth status.

Section 2.7. The protocol for RT-PCR is not completed: the extension step is missing.

Section 3. Results.

Rephrase the following subtitle Preparation of Lactation Activation Peptides from Oyster Hydrolysate.

As mentioned in the introductory section, English must be edited. Some phrases need to be reworded to make them easier to follow. Brake the long sentences in half to make them easier to read and understand.

Section 3.3.1. Related to the differences that were obtained between the groups mentioned in the body weight gain, could it not be due to the fact that some animals died? How was this controlled?

How it was verified that rat hypogalactia was induced? Please, mention in the text.

Section 4 and 5. Discussions and Conclusions.

Long sentences, hard to follow. Rewrite them.

Shorten the conclusions. 

If any, the limits of the study should be emphasized.

In conclusion, I recommend a substantial revision of the text and to make it a decent original article, I recommend adding missing data (study limits, final size data for each group with accurate reporting of the number of animals that died in each group). 

Reviewer 2 Report

Review

for the journal “Nutrients”

Research Paper “Lactation Activity and Mechanism of Milk-Protein Synthesis

from the Peptides of Oyster Hydrolysates”

Authors: Suhua Chen, Xiaoming Qin, Chaohua Zhang, Wenhong Cao, Huina Zheng and Haisheng Lin

  1. The authors investigated in detail the potential efficacy and mechanism of the active peptide UEC4-1 to improve postpartum hypogalactia in rats to provide a basis for studying the mechanism of lactation. This makes an impression in terms of novelty and methodological level and content of the work.
  2. Second page, second line. Please define: "AAs" play ...........?
  3. Table 1. Why were differences between mean group weights (g) not assessed?
  4. The quality of figures needs to be improved. It's complicated to read.
  5. 17. Literature source:  at the end of the text, two dots "107-114.." Need to be corrected.
  6. The results show that UEC4-1 significantly improves the lactation status and milk-protein synthesis of postpartum rats through the combined action of the PRL/AKT/STAT5 and mTOR/S6KB1 signaling pathways.
  7. The article is interesting, the conclusions are logical, but the above-mentioned corrections are recommended. I would suggest paying particular attention to the quality of the graphics. This is a technical issue as I have no comments on the content of the graphs.

Sincerely, reviewer.
